# Factors Associated with Postpartum Sexual Dysfunction in Spanish Women: A Cross-Sectional Study

**DOI:** 10.3390/jpm12060926

**Published:** 2022-06-02

**Authors:** Pedro Hidalgo-Lopezosa, Sandra Pérez-Marín, Andrea Jiménez-Ruz, Juan de la Cruz López-Carrasco, Ana María Cubero-Luna, Rubén García-Fernández, María Aurora Rodríguez-Borrego, Cristina Liébana-Presa, Pablo Jesús López-Soto

**Affiliations:** 1Instituto Maimónides de Investigación Biomédica de Córdoba (IMIBIC), 14004 Córdoba, Spain; z02pemas@uco.es (S.P.-M.); z72jirua@uco.es (A.J.-R.); juan.delacruz@imibic.org (J.d.l.C.L.-C.); anacuberoluna6@gmail.com (A.M.C.-L.); en1robom@uco.es (M.A.R.-B.); n82losop@uco.es (P.J.L.-S.); 2Departamento de Enfermería, Farmacología y Fisioterapia, Universidad de Córdoba, 14004 Córdoba, Spain; 3Hospital Universitario Reina Sofía de Córdoba, 14004 Córdoba, Spain; 4SALBIS Research Group, Departamento de Enfermería y Fisioterapia, Universidad de León, Campus de Ponferrada s/n, 24400 Ponferrada, Spain; rgarcf@unileon.es (R.G.-F.); cristina.liebana@unileon.es (C.L.-P.)

**Keywords:** sexual dysfunction, birth, postpartum period, perinatal factors, sex education

## Abstract

(1) Background: Female sexual dysfunction (FSD) has a high prevalence globally, and perinatal factors favor FSD, especially in the postpartum period. The aim was to determine the prevalence and factors influencing FSD in the postpartum period; (2) Methods: An observational study carried out in three primary care centers in southern Spain, with women in the postpartum period who had a single low-risk birth. One hundred and seventeen women answered the Female Sexual Function questionnaire during the 4th month postpartum, between January 2020 and December 2021. Sociodemographic, obstetric, neonatal variables and level of self-esteem were analyzed. A multiple logistic regression model was carried out; (3) Results: 78.4% had high level of self-esteem. FSD prevalence was 89.7%. Factors related to FSD were having an instrumental vaginal delivery, women with university studies, and prenatal preparation. Maternal age ≥ 35, multiparity, pathological processes in the child, a medium–low level of self-esteem and newborn weight were associated with disorders in some of domains of sexual function; (4) Conclusions: FSD is highly prevalent in the postpartum period and is associated with preventable factors. A preventive approach by health professionals to these factors is essential. Health services should implement postpartum follow-up programs, which may coincide in time and place with newborn follow-up programs.

## 1. Introduction

Female sexual dysfunction (FSD) encompasses any psychiatric problem or experiences in the individual and the couple which result in a loss of sexual desire, sexual arousal and orgasm and increased pain during intercourse [1]. This is a fairly common health problem which can negatively affect a woman’s quality of life [2]. Historically, research into FSD has been seriously neglected, and has only recently, over the last three decades, attracted more attention, unlike the studies into male sexual dysfunction, which date back to the early 20th century [3]. Globally, around 40% of women present some kind of sexual difficulty, although estimations of its prevalence threshold vary widely between 15% and 75% [4].

The origin of FSD is multifactorial, and the factors include chronic diseases such as diabetes, obstetric and gynecological diseases and surgery, as well as mental problems, amongst others. Of all these, the factors related to pregnancy, childbirth and postpartum are particularly important [5].

Female sexual function tends to decrease during pregnancy and remain low during the postpartum period [6]. However, reestablishing sexual relations during the postpartum period is a crucial factor for the couple’s relationship [7]. Postpartum FSD has a prevalence of 40% to 83% at 2 or 3 months after birth, levelling off at just over 60% at 6 months postpartum [8]. In addition to physical factors derived from the pregnancy and childbirth itself, it may also be affected by emotional and social factors, such as the couple’s own social adaptation to the new situation [9].

In addition, multiple factors arise during the postpartum that can trigger FSD, such as the fatigue produced by looking after a child, the stress of adapting to motherhood, breastfeeding, pain in the genital area, sleep disorders, the physical changes that the woman has experienced or psychological problems, such as postpartum depression [10].

Pain during intercourse, generally caused by the tears produced in childbirth [11] or from vaginal dryness as a result of high levels of progesterone [12], a lack of sexual desire and the inability to reach orgasm, worsen a woman’s sexual life, decreasing desire and sexual activity during the postpartum period [13]. Perineal trauma due to episiotomy or tears has also been linked to postpartum dyspareunia, which could lead to an increased risk of sexual dysfunction; however, there is at present insufficient evidence relating episiotomy to sexual function disorders [14].

There is a degree of controversy regarding the influence of the type of birth on FSD; although some studies associate vaginal birth and the accompanying perineal damage with an increased risk of FSD [15], other studies have not found a higher risk of FSD with vaginal birth compared to women who have had a caesarean section [16,17].

The main aim of this work, therefore, was to determine the prevalence of female sexual dysfunction in postpartum women in the south of Spain and to detect associated factors.

## 2. Materials and Methods

The study was approved by the Institutional Ethics Committee for Human Investigations of the province of Córdoba, Spain (Act. N° 280, ref. 414, 10 October 2018).

### 2.1. Design, Population, Sample and Study Period

We performed a descriptive, cross-sectional and analytical study with data collected in three primary care centers (health centers) in Córdoba (Spain) [18]. Characteristics, prevalence and factors related to female sexual dysfunction problems that appear in women during the postpartum period were studied. The sample was consecutive, including all the women who attended these three centers during the months of January and June 2020, with the inclusion criteria including women who: (i) had a single, low-risk vaginal or cesarean birth in one of the local public hospitals; (ii) attended the health center in the 4th month after birth; (iii) answered the validated questionnaire on female sexual function by Sanchez et al. (2004) [19]; and (iv) were over 18 years of age. Although 150 questionnaires were distributed, only 117 (78%) completed questionnaires were received.

### 2.2. Study Variables

The types of variables analyzed were: sociodemographic (age, educational level, employment); obstetric (parity, gestational age, prenatal preparation for birth, epidural analgesia, type of birth, episiotomy, vaginal tears, presence of stitches, pathology during pregnancy); neonatal (newborn (NB) weight, newborn pathology, admission to the neonatology unit, type of newborn feeding), and level of self-esteem using the Rosemberg scale (Atienza et al. 2000) [20], all of which variables were related to aspects of sexual function through the Female Sexual Function questionnaire.

### 2.3. Instruments and Procedure

The questionnaires, together with information on the study and the informed consent forms, were delivered and collected anonymously (in a double envelope) during the 4th month postpartum, on the occasion of the woman’s attendance at one of the scheduled consultations attended by the midwife.

All the women who met the inclusion criteria were given the questionnaires on female sexual function (Sánchez et al. 2004) [19] and the Rosemberg scale [20]. The Female Sexual Function Questionnaire consists of 14 questions on a Likert-type scale of 1 to 5 and is divided into the following domains: desire, arousal, lubrication, penetration, orgasm, sexual initiative, anticipatory anxiety, sexual communication and satisfaction. The first 6 questions refer to disorders of the different phases of the sexual response. Questions 7 and 8 assess relational aspects of sexual activity. Questions 9 and 10 assess sexual satisfaction. The remaining questions only describe aspects of interest about sexual activity. To diagnose sexual dysfunction only the first 6 questions are counted: Desire; Arousal; Lubrication; Orgasm; Problems with vaginal penetration; Anticipatory anxiety. Rosemberg’s Self-Esteem Scale (Atienza et al. 2000) [20] consists of 10 questions on a 1–4 Likert scale, and has a minimum score of 10 and a maximum of 40. The items included, half of which are phrased positively and half negatively, evaluate feelings of self-respect and self-acceptance.

### 2.4. Data Analysis

Data were obtained from the questionnaires and incorporated into a self-designed database for analysis. A descriptive and inferential analysis of the variables was carried out using the SPSS/PASW Statistic program V.25 (IBM Corp., Armonk, NY, USA). The qualitative variables were expressed in numbers (*n*) and percentages (%), and the quantitative variables in mean and standard deviation (SD). Hypothesis contrast tests were performed by applying the corresponding statistics according to the type of variable. The crude OR was calculated and subsequently adjusted by multiple logistic regression analysis (MLR) to measure the variables influencing sexual dysfunction. Having altered in at least four of the domains which are considered evaluators of sexual function (desire, arousal, lubrication, orgasm, penetration or anticipated anxiety) was considered as dependent variable in the MLR analysis. A 5% (*p* ≤ 0.05) α error was assumed, and the exact values of “*p*” were marked for each statistic.

## 3. Results

From the 150 surveys distributed during the data collection period, a total of 117 women participated (*N* = 117). The mean age was 32.8 years (±4.86), with a minimum age of 19 years and a maximum of 45 years. Of the women, 33.3% were 35 or older, while only 7.7% were 40 or older. Of the women, 35% had university education, 62.4% were primiparous and 65.8% of the women had received maternity education. The mean gestational age was 39.3 (±1.59) weeks of gestation and 63% had started labor spontaneously and 37% were induced births. Of the women, 76.9% had had a vaginal birth; of these, 59.8% had had a normal birth, while 17.1% had required operative vaginal birth. Episiotomy was performed in 51.3% and 35.9% of the women had suffered some type of tears, in most cases 1st and 2nd degree tears (90%). These affected primiparous women (34%) to a greater extent than multiparous women (16%). Of the women, 23% ended up having a cesarean birth. The complete sociodemographic, obstetric and neonatal data are shown in Table 1.

Most of the women stated having a high (78.4%) or normal/medium (14.7%) level of self-esteem, making a total of 93.1% who expressed a high or medium level of self-esteem. Only 6.9% reported having a low level of self-esteem. The mean score was 34.2 (±5.67) points, with a minimum score of 14 and a maximum of 40 on the Rosemberg scale.

Regarding sexual relations, 16.2% (*n* = 19) of the women declared they had engaged in no sexual activity during the last 4 weeks. Of the remaining 83.8% (*n* = 98) who had had sexual intercourse during the last 4 weeks, 18.1% stated that they had rarely or never felt the desire to engage in sexual activity with their partner, 30.3% stated that they had rarely or never been easily aroused, 17.2% that they did not become aroused after being touched during intercourse, 19.4% that they often, nearly always or always felt pain when their vagina was touched, 20.2% stated that penetration could rarely or never be carried out easily, 12.2% said that they often, nearly always or always felt afraid of the idea of engaging in sexual activity, 14.4% were rarely or never able to reach orgasm during sexual intercourse, and 11.2% reported having felt quite or extremely dissatisfied with sexual intercourse in the previous 4 weeks.

As regards the domains of sexual function, the most commonly affected were anticipatory anxiety (77.3% of the women), followed by problems related to penetration (70.4%), sexual initiative (64.3%), sexual desire (49.5%), arousal (39.4%), lubrication (38.8%), general sexual satisfaction (30.9%), satisfaction with sexual activity (28.6%), orgasm (25.8%), and to a lesser extent sexual communication (22.4%). Only 10.3% did not present any alteration and 3.2% presented alteration in only one aspect of sexual function; 21.6% had two altered aspects and 64.9% of the women declared that at least three aspects of sexual function had been altered. Of the women, 89.7% presented alterations (a score of 50% or below was equivalent to severe and/or moderate disorder) in at least one of the domains which are considered evaluators of sexual function: desire, arousal, lubrication, orgasm, penetration or anticipated anxiety. These results show that a risk of sexual dysfunction must be considered. The data are shown in Table 2.

Disorders in the different domains of sexual function were related to sociodemographic factors. For instance, maternal age was associated with a disorder in sexual initiative, as 87.5% of women aged 35 or over presented some type of disorder in this aspect compared to 51.5% of women under 35 years of age; the mother’s educational level was associated with problems in sexual arousal, thus women up to primary studies presented a lower degree of disorder than women with secondary and university studies; parity was related to communication problems, with multiparous women presenting a higher degree of disorder in this aspect than primiparous women; the type of delivery was associated with anticipated anxiety problems, so that women who had undergone a cesarean section presented a lower degree of disorder in this domain than women who had had a vaginal delivery; women with episiotomy had fewer problems with penetration; lower NB weight was related to problems with penetration; pathological NB processes were associated with a higher risk of impaired communication and sexual lubrication; and women with a low average level of self-esteem had a higher degree of impairment in satisfaction. These factors were not associated with disorders in the other domains of female sexual function. These data are shown in Table 3.

The MLR analysis (Table 4) showed that the factors that were associated with the risk of female sexual dysfunction were university studies (*p* = 0.015, OR: 4.45, 95% CI: 1.33–14.8), having an instrumental birth (*p* = 0.036, OR: 11.2, 95% CI: 1.16–107) and prenatal preparation (*p* = 0.019; OR: 0.23, 95% CI: 0.07–0.78).

## 4. Discussion

In the present study, FSD prevalence was high, and the factors related to FSD were having an instrumental vaginal delivery, women with university studies, and prenatal preparation. At the time of taking the questionnaire (4th month after delivery), 16.2% of the women declared they had not engaged in sexual activity during the last 4 weeks. These data are consistent with the study by Wallwiener et al., who found that 20% of women had not had sexual relations in the first three months postpartum [21]. Other recent studies report a significant decrease in sexual activity between 3 and 6 months postpartum, returning to normal at 12 months [22].

This study found a prevalence of 89.7% of sexual function disorders in the women. Other studies have recorded a prevalence of sexual dysfunction in women in the postpartum period of 64% [23], 30–60% [24], 68%16, 74% [25] or even as much as 86% [26]. In our study, the domains of sexual function most frequently affected were anticipatory anxiety (77%), followed by penetration problems (70%), sexual initiative (64%) and sexual desire (49.5%). Since being at risk of sexual dysfunction is defined as having a moderate or severe disorder in one of the factors considered evaluators of sexual activity (DEAS) [19], the percentage obtained here (89.7%) can be taken as the prevalence of sexual dysfunction in the study population. Khajehei et al. [23] concluded in their study on the prevalence and risk factors of FSD that the most commonly altered function was sexual desire (81%), followed by arousal (53%). A meta-analysis of 22 studies by Banaei et al. found that the prevalence of postpartum dyspareunia was 35% and this prevalence decreased with increasing duration of the postpartum [27].

There are multiple factors associated with the risk of FSD. In the present study, women with university studies had a higher risk of FSD. These data partly coincide with the study by Alp Yilmaz et al., who found a higher risk of sexual dysfunction in middle-class women with secondary education [25], although other authors have found a higher risk of FSD in women with low education levels [24]. On the other hand, women undergoing prenatal preparation had a lower risk of FSD in the present study and it could therefore be considered a protective factor. In this sense, some authors found this association when they implemented an educational package to a group of pregnant women. This intervention reduced sexual dysfunction by improving knowledge of and attitudes toward the physical and psychologic changes that occur during pregnancy among women attending routine prenatal care [28].

Another factor influencing sexual function was type of birth. Women who had an instrumental birth had higher FSD than women who had a non-instrumental one. Similarly, other authors found an increased risk of FSD in women with instrumented vaginal delivery due to the increased risk of injuries that may occur with this type of intervention during delivery [11,29]. Laganà et al. found that women with vaginal birth who had undergone an episiotomy had fewer problems in sexual function than women who had not, which could be due to the fact that the episiotomy reduces the risk of 3rd-4th degree tears [30], which are recognized as a cause of sexual dysfunction [10,11,31,32]. Despite the fact that many studies find no direct association between episiotomy and sexual dysfunction [26,31], they do consider 3rd–4th degree tears to be related to pelvic floor dysfunction [33]; however, there are many other studies which do find such an association [34,35,36]. In the present study, women who had undergone an episiotomy obtaining fewer problems in penetration than those who had not, however, in the adjusted RLM model, this variable did not appear as a factor associated with the risk of FSD in general. Some studies link vaginal birth with a higher risk of FSD, in that woman undergoing caesarean section had better scores on the Index of Female Sexual Function (IFSF) than women with vaginal delivery [15]. Nevertheless, in the study by Walwiever et al., women with a cesarean section showed a lower degree of sexual function [21], although a majority of studies have found no association between the type of birth and sexual dysfunction [16,17,37,38].

In this study, statistically significant differences were found in certain perinatal and sociodemographic variables when they were compared to some of the domains of sexual function, such as maternal age, where a higher percentage of women aged 35 or over presented a disorder in initiating sexual activity. In the study by Quoc Huy et al., women over the age of 30 had a higher risk of FSD, which could be attributable to the progressive fall in estrogens occurring in the gradual hormonal change which women experience as they get older [34]. In other studies, a higher risk of FSD was found in women aged 35 and over [29]. Parity is another variable analyzed in the present study, and despite not appearing as a factor associated with the risk of FSD in the RLM model, significant differences were found in terms of greater communication disorders in multiparous women; some authors have linked primiparity with a higher risk of FSD [15,34], although others have found a higher risk in multiparous women [26,29]. Although, in the present study, the level of self-esteem was not a variable associated with the risk of FSD according to the RLM model, we did find a higher percentage of alterations in general sexual satisfaction in women with a medium-low level of self-esteem. No studies were found to support this, although some studies were found in which the level of self-esteem was related to sexual arousal in young women [39].

The main strength of this work lies in the use of two questionnaires to address the issue in a more comprehensive way. This study has certain limitations. Firstly, the sample size was lower because many women declined to take the survey, and although 150 questionnaires were distributed only 117 women took part. Another limitation was the fact that data were missing from some of the questionnaires. In addition, for logistical reasons, the study was limited to three health centers in the city center. However, this research project has now been resumed and data are still being collected for a potential future publication that will address other aspects, as well as analyzing the results of the pandemic’s impact.

## 5. Conclusions

The results of the present study show a high prevalence of FSD in the postpartum period. The factors associated with FSD in this study were having an instrumental vaginal delivery, women with university studies, and prenatal preparation. In addition, maternal age ≥ 35, multiparity, pathological processes in the child, a medium–low level of self-esteem and NB weight were associated with disorder in some of the domains of sexual function.

The findings of this study stress the need to promote care for women to prevent FSD, starting at pregnancy and continuing into the postpartum period. Both the health professionals (nurses/midwives) who supervise the stages of pregnancy as well as health institutions should play their part in promoting this care. Similarly, further research into FSD should be encouraged to determine the factors that cause it.

## Figures and Tables

**Table 1 jpm-12-00926-t001:** Sociodemographic and obstetric characteristics of the women in the study (*N* = 117).

Variable	*n*	%
Age (years) *		32.8 (±4.86)
<35	78	66.7
≥35	39	33.3
Educational level		
Up to primary studies	38	32.5
Secondary studies	38	32.5
University studies	41	35.0
Employment		
Related to university studies	34	29.0
Related to secondary studies	31	26.5
Self-employed and services	38	32.5
Without work	14	12.0
Level of self-esteem		
High	91	78.4
Medium	17	14.7
Low	8	6.9
Parity		
Primiparous	73	62.4
Multiparous	44	37.6
Length of gestation (weeks) *	39.3 (±1.59)	
Pregnancy pathology		
Yes	31	26.5
No	86	84.5
Prenatal preparation		
Yes	77	65.8
No	40	34.2
Epidural		
Yes	99	84.6
No	18	15.4
Episiotomy		
Yes	60	51.3
No	57	48.7
Type of birth		
Normal	70	59.8
Instrumental	20	17.1
Cesarean	27	23.1
Vaginal tears		
Yes	42	35.9
No	75	64.1
NB in the neonatology unit		
Yes	21	17.9
No	96	82.1
NB feeding		
Breastfeeding	67	57.3
Artificial	23	19.7
Mixed	27	23
Newborn weight (g) *	3261 (±505)	

* Data are mean (±SD), NB: Newborn.

**Table 2 jpm-12-00926-t002:** Disorders in domains of sexual function in the postpartum period.

Domains of Sexual Function	Normal	Mild Disorder	Severe Disorder
	*n* (%)	*n* (%)	*n* (%)
Sexual desire	50 (50.5)	43 (43.4)	6 (6.1)
Arousal	60 (60.6)	32 (32.3)	7 (7.1)
Lubrication	60 (61.2)	20 (20.4)	18.4 (11.1)
Orgasm	72 (74.2)	15 (15.5)	10 (10.3)
Penetration problems	29 (29.6)	64 (65.3)	5 (5.1)
Anticipated anxiety	22 (22.7)	19 (19.6)	56 (57.7)
Initiating sexual activity	35 (35.7)	30 (30.6)	33 (33.7)
Sexual communication	76 (77.6)	10 (10.2)	12 (12.2)
Satisfaction with sexual activity	70 (71.4)	22 (22.4)	6 (6.2)
General sexual satisfaction	67 (69.1)	24 (24.7)	6 (6.2)

**Table 3 jpm-12-00926-t003:** Disorders in sexual function according to sociodemographic and perinatal factors.

Variable	Moderate and/or Severe Disorder *n* (%)	*p*	OR (IC 95%)
Age (years)	Initiating disorder		
<35	34 (51.5)		
≥35	28 (87.5)	0.001	6.58 (2.08–20.8)
Education level	Arousal		
Up to primary studies	39 (39.4)		
Secondary and university	32 (46.4)	0.031	0.35 (0.13–0.92)
Parity	Communication		
Primiparous	10 (16.7)		
Multiparous	13 (34.2)	0.046	2.60 (1.02–6.75)
Type of birth	anticipated anxiety		
Vaginal	60 (78.9)		
Cesarean	12 (57.1)	0.043	0.35 (0.12–0.99)
Episiotomy	Penetration problems		
Yes	29 (58)		
No	39 (81.3)	0.013	0.31 (0.12–0.79)
NB weight	Penetration problems		
≥3000 g	40 (61.5)		
<3000 g	28 (84.8)	0.018	3.50 (1.19–10.2)
NB pathology	Sexual communication		
Yes/No	4 (80)/19 (20.4)	0.010	15.5 (1.64–147)
	Lubrication		
Yes/No	5 (100)/34 (36.6)	0.008	1.14 (1.02–1.29)
Level of self-esteem	General sexual satisfaction		
Medium-low	12 (54.5)		
High	20 (26.7)	0.014	3.30 (1.23–8.81)

NB: Newborn.

**Table 4 jpm-12-00926-t004:** Factors associated with the risk of female sexual dysfunction.

Variable	Sexual Function Disorder Analysis			
	Univariant		Multivariant	
	*p*	Crude OR (IC 95%)	*p*	Adjusted OR (IC 95%) ^†^
Age (years)				
<35	Reference			
≥35	0.142	2.15 (0.77–5.97)		
Education level				
University studies	0.003	5.1 (1.75–15.3)	0.015	4.45 (1.33–14.86)
Type of birth				
Normal	0.255	3.47 (0.40–29.6)		
Instrumental	0.011	16.6 (1.90–30.5)	0.036	11.2 (1.16–107)
Cesarean	Reference			
Episiotomy	0.361	0.62 (0.22–1.71)		
Parity				
Primiparous	Reference			
Multiparous	0.771	1.16 (0.42–3.22)		
Vaginal tears	0.185	1.98 (0.72–5.10)		
NB weight (g) *	0.128	0.99 (0.99–1.01)		
NB pathology	0.257	2.94 (0.45–18.6)		
Breastfeeding	0.332	0.663 (0.30–1.66)		
Gestation age (weeks) *	0.287	0.85 (0.62–1.40)		
Prenatal preparation	0.004	0.21 (0.75–0.62)	0.019	0.23 (0.07–0.78)
E. Rosemberg (Self-esteem) *	0.133	0.94 (0.87–1.02)		
Employment				
University graduate	0.203	4.21 (0.38–44.0)		
Secondary school studies	0.678	1.39 (0.13–14.1)		
Self-employed/services	0.910	1.14 (0.11–11.1)		
Housewife	Reference			

* Data are mean (±SD), NB: Newborn; Hosmer Lemeshow: 0.537; ^†^ R^2^ Nagelkerke: 0.359.

## Data Availability

The data presented in this study are available on request from the corresponding author.

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
