# Peer review of "Factors Associated with Postpartum Sexual Dysfunction in Spanish Women: A Cross-Sectional Study"

_jpm, 2022, doi:10.3390/jpm12060926_

Round 1

Reviewer 1 Report

The presented study tackles an important issue of factors associated with postpartum sexual disfunction in Spanish women. I have red the article with a great interest. The study was conducted reliably with appropriate selection of tests. Overall, I think that this article should be published.  I have only three suggestions:

1. I suggest including the type of study inthe  Title and Methodology

 2. I suggest starting the Discussion from small resume of major findings.

3. I suggest including the information if you’re planning to prolong the study ( Results after 6 and 12 months might be very interesting).

Author Response

Reviewer 1 Notes:

Comments for authors:

The presented study tackles an important issue of factors associated with postpartum sexual disfunction in Spanish women. I have red the article with a great interest. The study was conducted reliably with appropriate selection of tests. Overall, I think that this article should be published.  I have only three suggestions:

  1. I suggest including the type of study in the Title and Methodology

Author´s answer:

The following sentence has been added in Design, population, sample and study period:

“We performed a descriptive, cross-sectional and analytical study with data collected in three primary care centers (health centers) in Córdoba (Spain)” (lines 75-76).

In the title it has been added the type of study: “Factors associated with postpartum sexual dysfunction in Spanish women: a cross-sectional study” (lines 2-3).

  1. I suggest starting the Discussion from small resume of major findings.

Author´s answer:

This sentence has been added at the beginning of Discussion section: “In the present study, the FSD prevalence was high, and the factors related to FSD were having an instrumental vaginal delivery, women with university studies, and prenatal preparation” (lines 200-202).

  1. I suggest including the information if you’re planning to prolong the study (Results after 6 and 12 months might be very interesting).

Author´s answer:

This sentence has been added at the end of Discussion section: “However, this research project has now been resumed and data are still being collected for a potential future publication that will address others aspects, as well as analysing the results by pandemic impact” (Lines 273-276).

Reviewer 2 Report

Line 86    Was BMI measured?  Might it be related to FSD?

Page 4   Some of the data are not lined up in columns and some data points to not include the decimal digit (i.e. 35 should probably be 35.0)

Line 159   3.2% rather than 3,2%

Line 190   Should the heading be General sexual dissatisfaction?

Line 216   "had a lower risk of FSD in the present study"

Line 222    "Another factor influencing..."

Line 262     "that data were missing from...."

Author Response

Reviewer 2 notes:

Line 86    Was BMI measured?  Might it be related to FSD?

Author´s answer:

BMI is an interesting variable that it might be related to FSD, howevwer, although we tried to include this variable, it was missing in most of the medical records.

Page 4   Some of the data are not lined up in columns and some data points to not include the decimal digit (i.e. 35 should probably be 35.0)

Author´s answer:

Data have been corrected in table 1

Line 159   3.2% rather than 3,2%

Author´s answer:

It has been corrected (line 167)

Line 190   Should the heading be General sexual dissatisfaction?

Author´s answer:

The name of the aspect of the affected sexual function is General sexual satisfaction, therefore it should be written in that form.

Line 216   "had a lower risk of FSD in the present study"

Author´s answer:

It has been corrected (line 226)

Line 222    "Another factor influencing..."

Author´s answer:

It has been corrected (line 232)

Line 262     "that data were missing from...."

Author´s answer:

It has been corrected (line 272)